# Centrosome Amplification Is a Potential Molecular Target in Paediatric Acute Lymphoblastic Leukemia

**DOI:** 10.3390/cancers15010154

**Published:** 2022-12-27

**Authors:** Meiyun Guo, Jenna Rever, Phuong N. U. Nguyen, Neha M. Akella, Gregor S. D. Reid, Christopher A. Maxwell

**Affiliations:** 1Department of Paediatrics, University of British Columbia, Vancouver, BC V6T 1Z4, Canada; 2Michael Cuccione Childhood Cancer Research Program, British Columbia Children’s Hospital Research, Vancouver, BC V5Z 4H4, Canada

**Keywords:** paediatric acute lymphoblastic leukemia, centrosome amplification, multipolar division, centrosome clustering, genomic instability

## Abstract

**Simple Summary:**

B-cell acute lymphoblastic leukemia (B-ALL) is the most common form of cancer in children. Current treatments deliver a high survival rate, but often cause harmful and enduring side effects. New treatments are needed to address this clinical challenge and reduce relapse and long-term effects in children. This study investigates the centrosome clustering pathway as a target for cancer treatments in children with B-ALL. Cancer cells often have enlarged or extra centrosomes and require the centrosome clustering pathway to progress through cell division successfully. Our data reveals that when the centrosome clustering pathway is disrupted in B-ALL cells it causes cell death and produces a population of damaged refractory cells. The refractory cells have markers that make them more visible to the immune system and are therefore more easily targeted by immune-based therapies.

**Abstract:**

Acute lymphoblastic leukemia (ALL) is the most common form of cancer in children, with most cases arising from fetal B cell precursor, termed B-ALL. Here, we use immunofluorescence analysis of B-ALL cells to identify centrosome amplification events that require the centrosome clustering pathway to successfully complete mitosis. Our data reveals that primary human B-ALL cells and immortal B-ALL cell lines from both human and mouse sources show defective bipolar spindle formation, abnormal mitotic progression, and cell death following treatment with centrosome clustering inhibitors (CCI). We demonstrate that CCI-refractory B-ALL cells exhibit markers for increased genomic instability, including DNA damage and micronuclei, as well as activation of the cyclic GMP–AMP synthase (cGAS)-nuclear factor kappa B (NF-κB) signalling pathway. Our analysis of cGAS knock-down B-ALL clones implicates cGAS in the sensitivity of B-ALL cells to CCI treatment. Due to its integral function and specificity to cancer cells, the centrosome clustering pathway presents a powerful molecular target for cancer treatment while mitigating the risk to healthy cells.

## 1. Introduction

Acute lymphoblastic leukemia (ALL) is a cancer of T- or B-lymphocytes and is characterized by the overproduction and accumulation of immature white blood cells in the bone marrow. It is the most common form of cancer in children, and accounts for 30% of childhood cancer cases [1]. Modern treatments have achieved overall survival rates approaching 90%, however 20% of patients experience a disease relapse [2]. In the paediatric population, the most common type of ALL is B-ALL, which originates from a fetal B-cell precursor cell of origin.

Genomic analysis of B-ALL, such as transcriptome sequencing [3,4,5] and optical genome mapping [6], has allowed for significant advancement in diagnostics and treatment through characterization of various B-ALL subtypes. Different subtypes of B-ALL are defined by specific genomic characteristics and are associated with distinct clinical outcomes. High hyperdiploid ALL is the most common subgroup in childhood B-ALL, and accounts for 25–30% of cases [7]. It is characterized by the non-random gain of chromosomes X, 4, 6, 10, 14, 17, 18, and 21 with a modal number of 51–67 chromosomes [7]. Although high hyperdiploid paediatric ALL generally has a favorable prognosis, about 20% of diagnosed patients will suffer a relapse and 10% will die from the disease [8]. Despite excellent prognosis at disease diagnosis, the outcomes for children that suffer a relapsed B-ALL remain very poor [9,10]. Moreover, patients that survive current treatments are at risk for adverse late-effects due to the damage current treatments cause to normal tissues [11,12]. Therefore, new therapies and molecular targets are required to treat relapsed B-ALL with reduced long-term effects in children.

Genome instability is a hallmark of cancer cells, which often results from chromosome segregation defects during mitosis and associates with an abnormal number of centrosomes in the produced daughter cells [13], termed centrosome amplification (CA). CA can cause the formation of a multipolar spindle, and commonly results in aneuploid cells or cell death. To avoid the lethal effects of multipolar division, cancer cells can group extra centrosomes to create a pseudo-bipolar spindle and progress through cytokinesis with fewer instances of chromosome segregation errors [14,15].

The grouping of extra centrosomes, termed centrosome clustering, utilizes molecular pathways that may be unique to cancer cells with CA, as the clustering process is not necessary during normal bipolar cell division. For example, kinesin family member C1 (KIFC1/HSET) is a minus-end directed kinesin-14 family member with a motor domain and an N-terminal microtubule-binding domain that can cross-link adjacent microtubules. Because of its minus-end motor activity [16], and the interaction with the centrosomal protein CDK5 regulatory subunit associated protein 2 (CEP215/CDK5RAP2) [17,18], KIFC1 can focus microtubule minus ends at spindle poles and promote the clustering of extra centrosomes. For this reason, silencing KIFC1 expression [19] or treatment with small molecules that inhibit KIFC1 activity, such as AZ82 [20], impairs the clustering process and results in mitotic cell death.

Alternatively, astral microtubules emanating from centrosomes can apply force against the cell cortex, and this force can result in the clustering of extra centrosomes [14]; for this reason, small molecule disruption of astral microtubule growth can inhibit the centrosome clustering process, such as via the inhibition of polo like kinase 1 (PLK1) activity, directly [21] or indirect upstream inhibition via Stattic [22,23]. Disruption of some of the components of the centrosome clustering molecular pathway have been shown to cause cancer-specific cell growth inhibition [13].

Paediatric B-ALL, and specifically high hyperdiploid B-ALL, may arise from an early abnormal mitotic event [24] and show cellular phenotypes that are consistent with ongoing mitotic instability including chromosome missegregation [25] and CA [26]. Here, we studied a cohort of primary paediatric B-ALL samples, normal stem cell samples, and immortal leukemia cell lines to gain insight into the prevalence of CA and the importance of centrosome clustering pathway for the viability of aneuploid B-ALL cells. Our data reveal CA to be prevalent in primary B-ALL cells with defective bipolar spindle formation resulting after treatment with centrosome clustering inhibitors (CCIs), which leads to abnormal mitotic progression and cell death. In response to CCI, the refractory or resistant B-ALL cells exhibit heightened genome instability, including DNA damage and micronuclei, and active cGAS-stimulator of interferon genes (STING) signalling with elevated expression of genes known to activate the innate immune system. These data reveal CA to be an effective molecular target for paediatric B-ALL, and small-molecule CCI as a potential conditioning therapy to be combined with immune-based treatments.

## 2. Materials and Methods

### 2.1. Primary Human Patient Samples

Primary patient samples were collected and provided from BC Children’s Hospital Biobank. All studies involving tissues or cell samples derived from human participants were approved by the UBC-affiliated Research Ethics Board (certificate #H18-01197). Leukemia cells were derived from bone marrow samples taken from consented B-ALL patients while normal primary stem cells were derived from non-involved bone marrow or peripheral blood samples taken from consented siblings or autologous donors. Clinical information of the patients is summarized in Appendix A.

### 2.2. Immortal Cell-Lines

Human B-ALL cell lines, including RS4;11, RCH-ACV and 380 cells, and a mouse B-ALL cell line, termed 289 cells, were a gift from Dr. Gregor Reid’s laboratory (BC Children’s Hospital Research Institute, Vancouver, BC, Canada). 289 cells were derived from a spontaneous leukemia arising in an Eμ-ret mouse [27,28]. Human mesenchymal stromal cells (MSCs) were a gift from Dr. Dario Campana’s laboratory (National University Cancer Institute, Singapore). These human MSCs were immortalized through the expression of telomerase reverse transcriptase protein (TERT) and green fluorescent protein (GFP) in primary bone marrow MSCs in Dr. Dario Campana’s laboratory, termed hTERT-MSCs. Species, age, karyotype, origin, and cytogenetics of immortal cell lines is summarized in Appendix A.

### 2.3. In Vitro Drug Profiling of Primary B-ALL Cells

The protocol described by Frismantas et al. [29] was adapted from their published 384-well format to a 96-well (Corning, Somerville, MA, USA) format used in this study. hTERT-MSCs were seeded at 3000 per well in 200 μL of RPMI-1640 medium containing 10% fetal bovine serum (FBS, Invitrogen, Carlsbad, CA, USA) and 1 μM hydrocortisone (Sigma, Burlington, MA, USA) 24 h prior to seeding with primary B-ALL or stem cells from peripheral blood or bone marrow. RPMI-1640 complete medium was removed before adding 5 × 10^4^ B-ALL cells, recovered from cryopreserved samples, in 200 μL of AIM-V medium. Both primary cells and hTERT-MSCs were incubated at 37 °C in a 5% (*v*/*v*) CO_2_ incubator for 48 h.

### 2.4. Cell Viability and Cytotoxicity Assay

The cell cytotoxicity assay kit was purchased from Abcam. Cell viability was assessed by treating cells with 1/5 volume of assay solution for 4 h at 37 °C. Absorbance at 570 and 605 nm wavelengths was measured with an EnSpireTM multilabel reader (PerkinElmer, Waltham, MA, USA). The background absorbance of the blank wells was subtracted from the values for the wells containing the cells. The ratio of OD570 to OD605 was used to determine the cell viability in each well.

### 2.5. Generation of 289 Cells Resistant to CCIs

289 cells were passaged 24 h before seeding in a 12-well plate (Corning). These cells were separately exposed to a half-maximal inhibitory concentration (IC50) dose of CCI (AZ82) or equivalent DMSO control. The drug, or DMSO, was refreshed every 2 days for 2 weeks. Then, the drug concentration was increased by 33% (1.33 fold) for an additional 2 weeks. This cycle was repeated 6 times until a population of 289 cells that were resistant to lethal doses of the drug was generated. These cell lines were termed 289r (DMSO) or 289r (AZ82).

### 2.6. Generation of 289 cGAS Knock-Down Clones Using CRISPR-Cas9

Guide RNA (gRNA) oligonucleotides were designed using the University of California San Francisco Genome Browser and ordered from Integrated DNA Technologies. Overhangs for ligation to BbsI sites in plasmid pSpCas9(BB)-2A-GFP (Addgene PX458) were added to the 5′ and 3′ ends of the forward and reverse oligos, as described in Ran et al., 2013 [30]. Each gRNA oligonucleotide pair was incubated at 95 °C for 10 min. pX458 was digested with BbsI for 1.5 h at 37 °C. The digested pX458 was incubated with annealed oligonucleotides and T4 DNA ligase (Thermofisher, Waltham, MA, USA) for 1.5 h at 37 °C. After ligation, PX458 was transformed into DH5α cells (Thermofisher, Waltham, MA, USA), and single colonies were selected for Sanger sequencing to confirm gRNA insertion.

PX458 with the desired gRNA insertion was transformed into 289 cells using electroporation (NEPA21 Electroporator). The next day GFP-positive transfected cells were sorted as single cells into a 96 well plate using fluorescence-activated cell sorting (FACS). 2 weeks after single cell sorting, genomic DNA was extracted with the DNeasy kit (Qiagen, Hilden, Germany), and used as a template for PCR amplification of the clustered regularly interspaced short palindromic repeats (CRISPR)-Cas9 targeted region of cGAS. PCR conditions were: PCR cycling conditions were: 95 °C for 5 min, followed by 35 amplification cycles at 95 °C for 60 s, 58 °C for 30 s, and 72 °C for 1 min. PCR product was purified using the QIAquick PCR kit, and analyzed with sanger sequencing to confirm putative clones with the desired genetic editing.

### 2.7. Immunofluorescence

Cells were seeded on poly-L-lysine coated coverslips and fixed 4% PFA for 10 min, then permeabilized with ice-cold methanol for 10 min at −20 °C. Cells were blocked in PBS with 0.2% Triton X-100 and 3% BSA for 1 h at room temperature. Antibodies were diluted in PBS with 0.2% Triton X-100 and 3% BSA. Primary antibodies were diluted and incubated with coverslips overnight at 4 °C. Cells were then washed three times in PBS. The coverslips were incubated with diluted secondary antibodies at room temperature for 1 h in the dark. Coverslips were washed three times in PBS and mounted with ProLong Gold Antifade Reagent containing DAPI (Invitrogen).

### 2.8. Confocal Microscopy and Image Acquisition

Fixed cells were imaged using the Fluoview software (Olympus, Tokyo, Japan) connected to the Olympus Fluoview FV10i confocal microscope. Image stacks of 20 optical sections with a spacing of 0.5 µm through the cell volume were taken using a 60 × 1.2 NA oil objective. ImageJ v1.46j (National Institute of Health) was used to generate maximum intensity projection of the fluorescent channels.

### 2.9. High-Content Imaging

Cells were seeded in a 96-well plate (Corning) and placed in the chamber of the high content image analysis system (ImageXpress Micro XL). Images were taken through a 40X 0.75 NA dry objective on the ImageXpress Micro XL epifluorescence microscope (Molecular Devices Inc., San Jose, CA, USA) controlled by the MetaXpress 5.0.2.0 software (Molecular Devices Inc.). For the analysis of the proportion of living cells, images were taken once per site using 50-ms exposures, 2 × 2 binned resolution, with 100% of full lamp intensity for each channel, and 25 optical sections spaced 500 μm apart. For analysis of micronuclei and gamma-H2A histone family member X (γ-H2AX), images were taken once per site using 50-ms exposures, 2 × 2 binned resolution, with 100% of full lamp intensity for each channel, and 100 optical sections spaced 700 μm apart. Post-acquisition processing of images was performed using MetaXpress offline.

### 2.10. TaqMan Mouse Immune Array

RNA was extracted from 289r (DMSO) cells and 289r (AZ82) cells using AllPrep DNA/RNA mini kit (Qiagen). Reverse transcription PCR (RT-PCR) was carried out using the SuperScript IV VILO Master Mix (Thermo Fisher, Waltham, MA, USA) kit. RT-PCR conditions were: 25 °C for 10 min, 50 °C for 10 min, and 85 °C for 5 min. cDNA quantity was determined by the QuantiFluor dsDNA System (Promega, Madison, WI, USA). Quantitative PCR (qPCR) reactions were run on TaqMan Mouse Immune Array (Thermo Fisher) assay, using StepOnePlus system (Applied Biosystems, Waltham, MA, USA). qPCR cycling conditions were 50 °C for 2 min and 95 °C for 20 s, followed by 40 cycles at 95 °C for 1 s and at 60 °C for 20 s. Analysis of qPCR results was done using the cycle threshold (Ct) values normalized to 18S ribosomal RNA (18S rRNA). Genes are listed in Appendix A.

### 2.11. qPCR Analysis of cGAS Knock-Down Clones

RNA was extracted from 289 knock-down clones using AllPrep DNA/RNA mini kit (Qiagen). RT-PCR was carried out using the SuperScript IV VILO Master Mix (Thermo Fisher, Waltham, MA, USA) kit. RT-PCR conditions were: 25 °C for 10 min, 50 °C for 10 min, and 85 °C for 5 min. qPCR reactions were run with SyberGreen mastermix (Thermo Fisher, Waltham, MA, USA), using the StepOnePlus system (Applied Biosystems, Waltham, MA, USA). qPCR cycling conditions were 50 °C for 2 min and 95 °C for 20 s, followed by 40 cycles at 95 °C for 1 s and at 60 °C for 20 s. Analysis of qPCR results was done using the Ct values normalized to glyceraldehyde-3-phosphate dehydrogenase (GAPDH).

### 2.12. Statistics

Statistical analysis was performed using GraphPad Prism. Data were expressed as mean ± standard deviation (SD) or as mean ± standard error of mean (SEM) as indicated in each figure. Statistical analysis was performed by One-way analysis of variance (ANOVA) as indicated in each figure, except for Figures 5a and 7d, which were analyzed with a two-tailed unpaired *t*-test. Pearson r was used to determine correlation between CA and IC50 in heatmaps for Figures 2b and 4b, and the correlation between CA and % micronuclei in Figure 5b. The results were considered statistically significant at *p* < 0.05.

## 3. Results

### 3.1. Centrosome Clustering Is Required for the Viability of B-ALL Cell Lines

Defective chromosome segregation can result in aneuploidy and an abnormal number of centrosomes in daughter cells. We investigated the frequency and type of CA in three human B-ALL cell lines and one mouse B-ALL cell line (Appendix A) using immunofluorescence to stain different components of the centrosome, including the centriole (centriolar coiled coil protein 110 (CP110)), pericentriolar material (γ-tubulin)) and emanating microtubules (β-tubulin) (Figure 1a), in both interphase (i) and mitotic (m) cells. CA was defined as ≥3 centrosomes, or centrosomes larger than 4 μm in diameter. Immunofluorescence analysis revealed 40.7% of mitotic 289 B-ALL cells exhibited CA, with lower frequencies measured in RCH-ACV (32%), 380 (16.7%), and RS;411 (19.2%) cells (Figure 1b). Thus, CA is present in immortal B-ALL cell lines but does not appear to be ubiquitous and varies in frequency and type of abnormalities.

Cancer cells must cluster extra centrosomes during cell division to form a pseudo-bipolar mitotic spindle and avoid the lethal effects of multipolar cell division. To evaluate the sensitivity of B-ALL cells to CCIs, we treated four immortal B-ALL cell lines for 48 h with six graded doses of CCIs AZ82 and Stattic, in addition to the chemotherapy drugs Doxorubicin and Paclitaxel (Appendix A). As shown by the respective dose response curves (Figure 2a), B-ALL cell lines were sensitive to both CCIs and chemotherapy drugs (Figure 2a). Furthermore, higher frequency of CA in the cell lines correlated with increased sensitivity (lower dose) to both CCIs and Doxorubicin (Figure 2b). Interestingly, the opposite was true for sensitivity to Paclitaxel, which was anticorrelated to the frequency of CA in the cell lines. To investigate if AZ82 and Stattic induce centrosome declustering, 289 cells were treated for five hours with either AZ82 or Stattic. We evaluated the frequency of multipolar spindles with immunofluorescence (Figure 2c,d) and found it was significantly higher in both CCI treated samples (37%, 44%) compared to the DMSO control (3%). These data indicate that the centrosome clustering pathway is required for the survival of B-ALL cell lines, and increased levels of CA confer an increased sensitivity to CCIs.

### 3.2. The Efficacy of CCIs Correlates with the Levels of CA Measured in Primary B-ALL Patient Samples

To query whether CA is also prevalent in primary B-ALL cells, we first examined the frequency of CA in a cohort of primary paediatric B-ALL samples (n = 30; age: 30 months–200 months) and control stem cell samples sourced from donors of similar age (n = 6; age: 18 months–200 months) (Appendix A). Immunofluorescence analysis revealed CA frequency across the cohort of paediatric B-ALL samples ranged between 23% and 61% (Figure 3a), significantly higher than in stem cell control samples (<10% of cells), and similar to the levels observed in B-ALL cell lines (Figure 3b). CA levels were significantly higher in the aneuploid (high hyperdiploid or hypodiploid) B-ALL samples than in the euploid (translocation-positive or diploid) B-ALL samples, but CA levels did not significantly vary based on patient sex, disease stage, or disease risk estimates (Figure 3c).

To measure the efficacy of CCI against primary paediatric B-ALL cells, we first cultured primary B-ALL cells on hTERT-MSCs, which maintained viability through 48 h significantly better than adherent culture alone or adherent culture with CD40 Ig stimulation (Appendix A). In these co-cultures, we optimized image-based measurements of drug sensitivity by exposing primary B-ALL cells separately to two CCIs (AZ82 and Stattic) (Appendix A) and measured viable cells after 48 h through high-content image analysis or manual counting with significant concordance in the results (AZ82: r2 = 0.99, *p* < 0.0001; Stattic: r2 = 0.98, *p* < 0.0001) (Appendix A). Finally, we tested the selected inhibitors against two independent aliquots of four primary samples and measured highly correlated IC50 and %CA values (Appendix A) indicating reproducibility of measurements across aliquots. However, in these pilot experiments we noted the tested primary B-ALL cells were not more sensitive to Stattic (Appendix A) and, consequently, we excluded Stattic from subsequent measurements.

We expanded our analysis of CCIs to include a representative cohort of primary B-ALL samples, with 4 stem cell control samples and 13 B-ALL samples, and measured IC50 values following exposure to Doxorubicin, AZ82, or DMSO control (Figure 4a). When the level of CA in mitotic cells from primary samples and drug sensitivity (IC50) were plotted as a heatmap, hierarchical clustering of the data separated the four primary samples derived from non-involved bone marrows (Figure 4b). For primary patient B-ALL samples, sensitivities to AZ82 and Doxorubicin significantly correlated with the frequency of CA (Figure 4b), similar to our findings from immortal B-ALL cell lines (Figure 2b). Overall, these studies indicate that CCIs are effective at reducing the proliferation of paediatric B-ALL cells, and responses are proportional to the frequency of CA in the treated cancer cell population.

### 3.3. B-ALL Cells Refractory to CCIs Display Markers for Genome Instability and Activate cGAS-NF κB Proinflammatory Signaling

Tumor cells generally respond to targeted therapies but often a population of cells becomes refractory and, in the case of CCI-treated populations, may exhibit genomic instability. So, we examined the nucleus architecture in B-ALL cells that survived 48 h after drug treatment via the DNA staining dye CyQuant and measured nuclear atypia and defects in nuclear architecture, focusing on the frequency of small extranuclear bodies termed micronuclei. In DMSO-treated primary cells, we measured no significant difference in micronuclei frequency between normal stem cells and primary leukemia samples. However, the frequency of micronuclei in primary B-ALL cells treated with the CCI AZ82, or the chemotherapy doxorubicin, was significantly higher than that observed in drug-treated non-involved bone marrow stem cells (Figure 5a). Heatmap plots showed that primary cells with high levels of CA were not only more sensitive to CCIs, as determined by lower IC50 levels (Figure 2e), but also created a population of refractory cells with an elevated occurrence of micronuclei (Figure 5b), which is consistent with the induction of multipolar spindles and mitotic instability in this population.

To model the responses of B-ALL cells to the CCI AZ82, we created 289 sublines that selectively resist AZ82 (termed 289r (AZ82) cells) (Figure 6a) (Appendix A) and measured the frequency of micronuclei in these populations. As a positive control to induce micronuclei, parental 289 cells were subjected to graded doses of X-radiation. In these control experiments, we observed dose-dependent increases in the frequency of micronuclei in the population of parental 289 cells. Similarly, we observed a significantly increased frequency of micronuclei in 289r (AZ82) cells relative to DMSO-treated 289r (DMSO) control cells (Figure 6b). We then analyzed the level of γ-H2AX foci in 289r (AZ82) cells as micronuclei are known to be particularly unstable and prone to ongoing DNA damage. Compared to the DMSO-treated 289r (DMSO) control cells, we found the level of γ-H2AX was significantly elevated in 289r (AZ82) cells and in parental 289 cells exposed to graded levels of radiation (Figure 6c). Moreover, we used immunofluorescence to localize the cytoplasmic enzyme cGAS to the majority of micronuclei (Figure 6d), and we also observed an elevated expression of proinflammatory transcripts (elevated C-X-C motif chemokine ligand 10 (CXCL10), C-C motif chemokine ligand 3 (CCL3), reduced suppressor of cytokine signaling 1 (SOCS1)) (Figure 6e) in 289r (AZ82) cells. Thus, 289r (AZ82) cells, which are refractory to CCI, show evidence of elevated genetic instability (γ-H2AX foci) and genomic instability (micronuclei frequency), with activation of cGAS-NF-kB signaling and pro-inflammatory gene expression.

### 3.4. cGAS Is Necessary for 289 B-ALL Cell Responses to CCI

To evaluate the necessity for cGAS in the response of 289 B-ALL cells to CCIs, we used CRISPR-Cas 9 to generate three cGAS knock down (KD) clones targeting exon 2 (KD.38, KD. 32, and KD.07) selecting one clone showing a parental phenotype (KD.32), and two clones showing a null (KD.07) or near null (KD.38) phenotype. After detecting gene alterations with Sanger sequencing (Figure 7a), we X-irradiated three putative KD clones and parental 289 cells to functionally validate the loss of cGAS-STING proinflammatory signaling. Twenty-four hours after sham treatment or 1 Gy X-irradiation, we isolated mRNA and used qPCR to quantify the relative expression of cGAS, STING, and interferon alpha 1 (IFN-α1). In relation to sham-treated control cells, 1 Gy X-irradiated parental 289 cells significantly increased the expression of cGAS and IFN-α1, without similar increases in STING or GAPDH transcripts, when measured 24 h after irradiation. However, KD.07 and KD.38 had an absent or significantly lower increase in cGAS and IFNα expression, indicating the ablation of cGAS response to dsDNA in the cytoplasm caused by irradiation (Figure 7b). Next, we measured the necessity for cGAS expression during the proliferation of 289 cells in standard culture or when stressed through treatment with a CCI. Analysis of cell proliferation over 72 h revealed no significant difference between 289 parental and cGAS KD lines (Figure 7c), indicating the reduction of cGAS has no effect on cell survival under normal growth conditions. However, reduction of cGAS was protective when cells were treated with a CCI. That is, treatment with 0.5 μM AZ82 for 24 h resulted in a 80% reduction in viability for 289 parental cells and KD.32 cells (Figure 7d). However, sensitivity to AZ82 was greatly reduced in KD.07 cells, with KD.38 cells showing a middle phenotype (Figure 7d). Thus, cGAS appears to be needed for tumor suppressive responses after exposure to the CCI, AZ82.

## 4. Discussion

The clustering of supernumerary centrosomes is an essential molecular pathway found in cancer cells with CA to enable pseudo-bipolar cell division. This study reveals, through analysis of immortal cell lines and primary samples, that CA is prevalent in paediatric B-ALL, and the centrosome clustering pathway is required for cell viability. From analyses of four B-ALL cell lines of human (RCH:ACV, RS4;11, 380) and mouse (289) origin, CA is shown to be common but not ubiquitous in frequency (Figure 1). This discrepancy could be attributed to the parameters defined for identification of CA in this study, as centrosome diameter was used for classification as opposed to total centrosome volume. We identified centrosomes as composed of a CP110-positive centriole, encapsulated by gamma-tubulin-positive pericentriolar material, and nucleating beta-tubulin-positive microtubules, although alternative antibodies such as pericentrin have also been previously used. Furthermore, this analysis was limited by the detection sensitivity of immunofluorescence assays. Clonal heterogeneity could also play a role in CA variance between cell lines, depending on the dominant clone’s centrosome morphology. Nonetheless, the frequency of CA observed in the cell lines was supported by a reciprocal induction of multipolar spindles by CCIs (Figure 2) and reflected in the frequency of CA in patient samples (Figure 3).

In addition to being prevalent in B-ALL cells, our data shows CA is positively correlated to cellular response to CCIs in both human and mouse cell lines (Figure 2), and primary samples (Figure 3). The specificity of CCI sensitivity to B-ALL cells with CA, in comparison to stem cell controls, indicates a mitigation of risk to healthy cells during treatment. Although cell lines were found to be sensitive to both Stattic and AZ82, preliminary experiments revealed primary B-ALL samples were not sensitive to Stattic (Appendix A). Notably the 289 cell line, derived from spontaneous primary leukemic cells arising in Eμ-ret mice, was significantly more sensitive to Stattic in comparison to the other cell lines. This distinctive response may be attributed to the activation of signal transducer and activator of transcription 3 (STAT3) by Eμ-ret fusions [31,32], and the specific sensitivity of 289 cells to Stattic. We also observed a relative insensitivity to Paclitaxel for immortal cell lines with elevated levels of CA. It is likely that this response reflects the increased microtubule nucleation capacity observed in cells with CA [33], which suggests that measurement of CA may be predictive of responses to Paclitaxel in use for multiple cancers.

CA occurs in almost all cancer types and has been found to be associated with important prognostic indicators such as stage, grade, and risk [34,35]. For example, CA is common in the adult blood cancer multiple myeloma [36], wherein CA correlates with the elevated expression of centrosome gene products [37]; indeed, a gene expression-based centrosome index was developed in myeloma [38] and may serve to complement our immunofluorescence-based analysis of CA in paediatric B-ALL. Furthermore, a previous study of 50 B-ALL patients, using similar methods, found 72% of patients exhibited abnormal centrosome phenotypes, lending further support to the prevalence of CA in B-ALL. Given the correlation presented between AZ82 sensitivity and frequency of CA in B-ALL, this study proposes the utility of CA frequency as an effective companion diagnostic to inform application of CCI’s in B-ALL treatment plans.

While genome instability is a hallmark of cancer and a proposed enabling characteristic for relapse [39], there is an accompanying detriment to tumor viability. This study demonstrates mouse 289 B-ALL cells refractory to CCI’s have increased genomic instability in the form of micronuclei which activate the cGAS-STING pathway, therefore increasing cell immunogenicity. cGAS has been shown to act in a tumor suppressive manner through increased immune surveillance, promoting cell senescence, and contributing to apoptotic cell death in cancer cells [40]. Taken together, our data presents a utility for CCI’s in combination with immune modulators for treatment of B-ALL with high frequency of CA.

## 5. Conclusions

Disease relapse and long-term side effects remain significant clinical challenges in paediatric B-ALL. This study investigates CA and the centrosome clustering pathway as a powerful molecular target for B-ALL treatment. Here, we find that CA is prevalent in primary and immortal B-ALL samples, sourced from humans or mice, and the frequency of CA within B-ALL samples correlates to the sensitivity of these samples to CCIs. This is reflected in lower IC50 doses and augmented instability measured in populations with higher CA frequency. We find that CCI-refractory populations locate cGAS to micronuclei and express pro-inflammatory gene products. Finally, cGAS knock-down in B-ALL cells with a high level of CA is protective against the actions of a CCI indicating the cGAS pathway may be required for the optimal sensitivity of B-ALL cells to the inhibition of centrosome clustering.

## Figures and Tables

**Figure 1 cancers-15-00154-f001:**
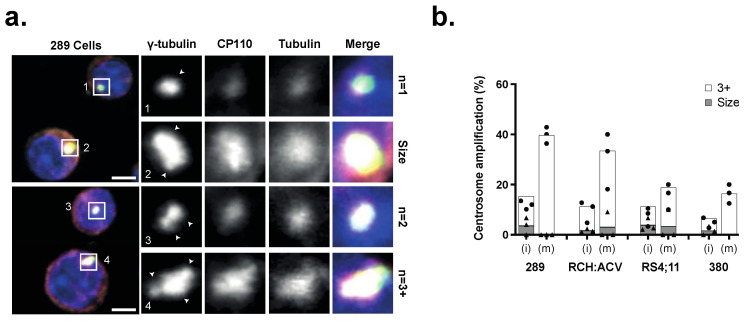
(**a**) Immunofluorescence detection of centrosomes with γ-tubulin, CP110, β-tubulin, and DAPI in mouse 289 cells. Scale bar = 4 μm. Arrows indicate individual centrosomes. (**b**) Percentage of cells displaying CA in interphase (i) and mitotic cells (m) in four B-ALL cell lines. Cells with either supernumerary centrosomes (≥3) (white) or large centrosomes (≥4 μm^2^) (grey) were classified as CA. (*n* = 3) experiments indicated by dots (white, 3+) or triangles (grey, size), *n* = 20 cells/experiment).

**Figure 2 cancers-15-00154-f002:**
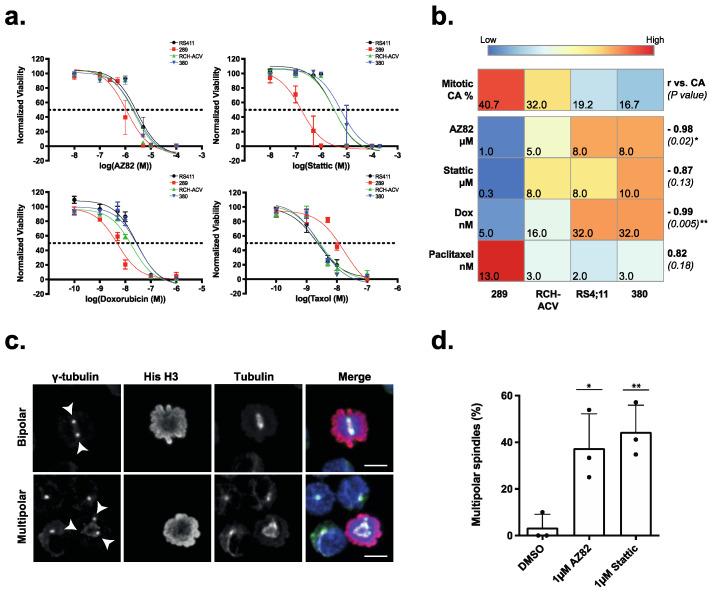
(**a**) Cell viability normalized to DMSO control in four B-ALL cell lines treated for 48 h with serial dilutions of CCIs (AZ82, Stattic) or chemotherapy drugs (Doxorubicin, Paclitaxel). IC50 value is indicated by the dashed line. (mean ± SEM, *n* = 3 experiments). (**b**) Correlation between % CA and IC50 from CCIs (AZ82, Stattic) or chemotherapy drugs (Doxorubicin, Paclitaxel) for four B-ALL cell lines. Heatmap is scaled by row Z-score, with Pearson r (vs. CA) values (bold) and the *p*-value (in parentheses) on the right side (Pearson r, * *p* < 0.05, ** *p* < 0.01). (**c**) Detection of bipolar and multipolar cell division in mitotic 289 mouse cells stained with γ-tubulin, Histone-H3, β-tubulin, and DAPI. Cells were incubated with 1 μM AZ82 for 5 h. Scale Bar = 5 μm. (**d**) Frequency of mitotic 289 cells with multipolar spindles incubated with 1 μM AZ82, 1 μM Stattic, or 1 μM DMSO for 5 h. (mean ± SD, *n*=3 experiments indicated by dots, >20 cells per bar, * *p* < 0.05, ** *p* < 0.01, One-way ANOVA).

**Figure 3 cancers-15-00154-f003:**
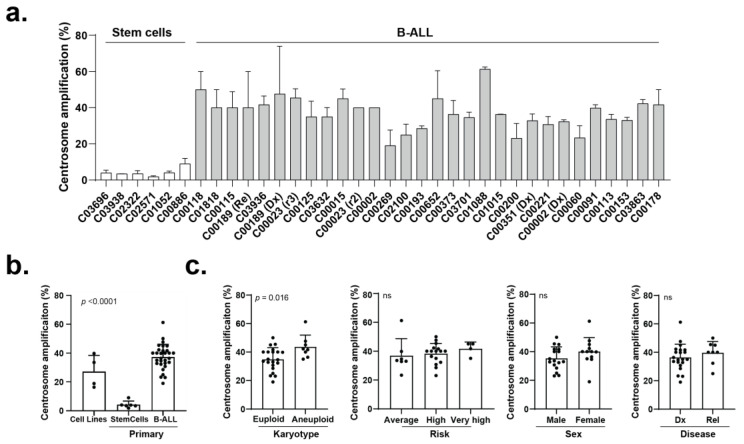
(**a**) Frequency of CA in 30 primary B-ALL samples and six primary bone marrow stem cell samples. (mean ± SD, >20 cells per bar). (**b**) Frequency of CA in 4 cell lines, 30 primary B-ALL samples, and 6 primary bone marrow stem cell samples. (mean ± SD, *p* = <0.0001, One-way ANOVA). (**c**) Frequency of CA in primary B-ALL samples comparing karyotype (euploid and aneuploid), risk level (average, high, very high), sex (male and female), and stage (initial diagnosis (Dx) and relapse (Rel)). (mean ± SD, ns = not significant, One-way ANOVA, *p*-value indicated on graph).

**Figure 4 cancers-15-00154-f004:**
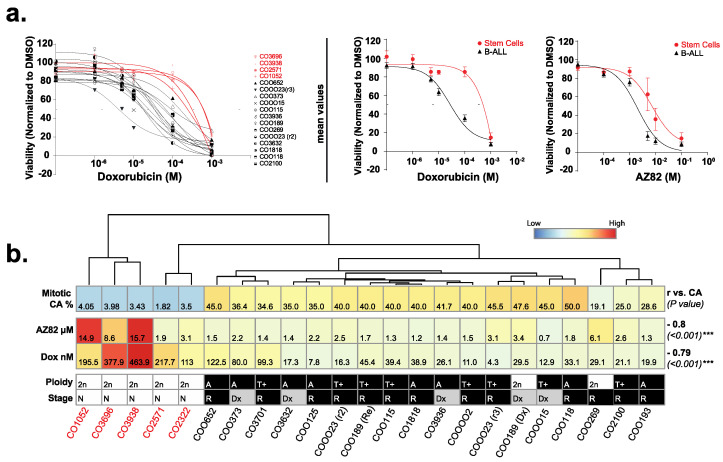
(**a**) Cell viability normalized to DMSO control in 18 primary B-ALL samples and 5 primary bone marrow stem cell samples treated for 48 h with serial dilutions of AZ82 or Doxorubicin. (mean ± SEM). (**b**) Correlation between %CA and IC50 from AZ82 or Doxorubicin for 18 primary B-ALL samples and 5 primary bone marrow stem cell samples. Heatmap columns are ordered by hierarchical clustering based on average linkage, and rows are scaled by Z-score. Pearson r (vs. CA) values (bold) and the *p*-value (in parentheses) are on the right side (Pearson r, *** *p* < 0.001). Primary sample ploidy (2n = Euploid, A = aneuploid, T+ = translocation aneuploid), and stage (N = none, Dx = initial diagnosis, R = relapse) are indicated below the heatmap.

**Figure 5 cancers-15-00154-f005:**
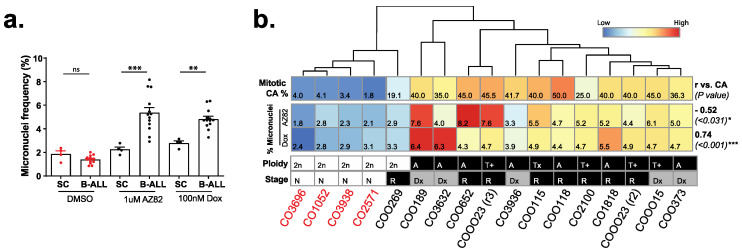
(**a**) Frequency of micronuclei in 4 primary bone marrow stem cell (SC) samples and 13 primary B-ALL samples incubated with 1 μM AZ82, 100 nM Doxorubicin, or DMSO for 48 h. (mean ± SEM, unpaired two-tailed *t*-test, ** *p* < 0.01, *** *p* < 0.001, ns = not significant). (**b**) Correlation between % CA and % micronuclei for 13 primary B-ALL samples and four primary bone marrow stem cell samples incubated in 1 μM AZ82 or 100nM Doxorubicin. Heatmap columns are ordered by hierarchical clustering based on average linkage, and rows are scaled by Z-score. Pearson r (vs. CA) values (bold) and the *p*-value (in parentheses) are on the right side (Pearson r, * *p* < 0.05, *** *p* < 0.001). Primary sample ploidy (2n = Euploid, A = aneuploid, T+ = translocation aneuploid), and stage (N = none, Dx = initial diagnosis, R = relapse) are indicated below the heatmap.

**Figure 6 cancers-15-00154-f006:**
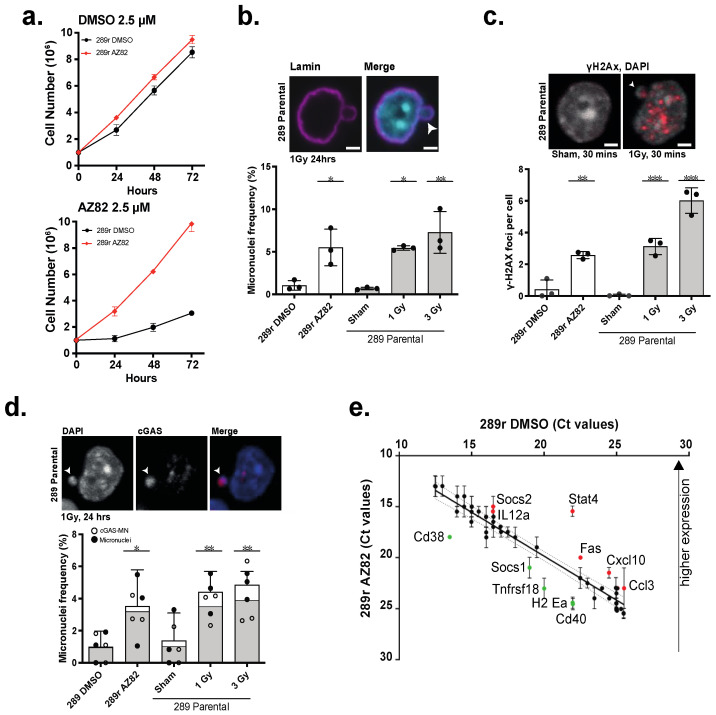
(**a**) Cell proliferation of 289r AZ82 and 289r DMSO cells incubated with 2.5 μM AZ82 or 2.5 μM DMSO over 72 h. (mean ± SEM, n = 3 experiments). (**b**) Micronuclei frequency in 289r AZ82 and 289r DMSO cells, and 289 parental cells 24 h after three X-irradiation conditions (sham, 1 Gy, 3 Gy). (mean ± SEM, n = 3 experiments, >500 cells per bar, * *p* < 0.05, ** *p* < 0.01, One-way ANOVA). Image shows 289 parental cell line 24 h post 1 Gy X-irradiation stained with lamin and DAPI. Arrow indicates a micronucleus. Scale bar = 2 μm. (**c**) γ-H2Ax foci per cell in 289r AZ82 and 289r DMSO cells, and 289 parental cells 24 h after three X-irradiation conditions (sham, 1 Gy, 3 Gy). (mean ± SEM, n = 3 experiments, >500 cells per bar, ** *p* < 0.01, *** *p* < 0.001, One-way ANOVA). Image shows 289 parental cell line 30 min post 1 Gy X-irradiation or sham treatment stained with γ-H2Ax and DAPI. Arrow indicates a micronucleus. Scale bar = 2 μm. (**d**) Frequency of micronuclei (white) and micronuclei co-localized with cGAS (grey) per cell in 289r AZ82 and 289r DMSO cells, and 289 parental cells 24 h after three X-irradiation conditions (sham, 1 Gy, 3 Gy). (mean ± SEM, n = 3 experiments, >500 cells per bar, * *p* < 0.05, ** *p* < 0.01, One-way ANOVA). Image shows 289 parental cell line 30 min post 1 Gy X-irradiation or sham treatment stained with cGAS and DAPI. Arrow indicates a micronucleus. Scale bar = 2 μm. (**e**) Ct values for expression of 48 mouse immune genes between 289r AZ82 and 289r DMSO control cell lines. Each dot represents a gene and dotted lines show the associated 95% confidence interval. Red or green color indicate the up- or down-regulated genes, respectively in 289r (AZ82) cells. (n = 2 experiments, mean ± SEM).

**Figure 7 cancers-15-00154-f007:**
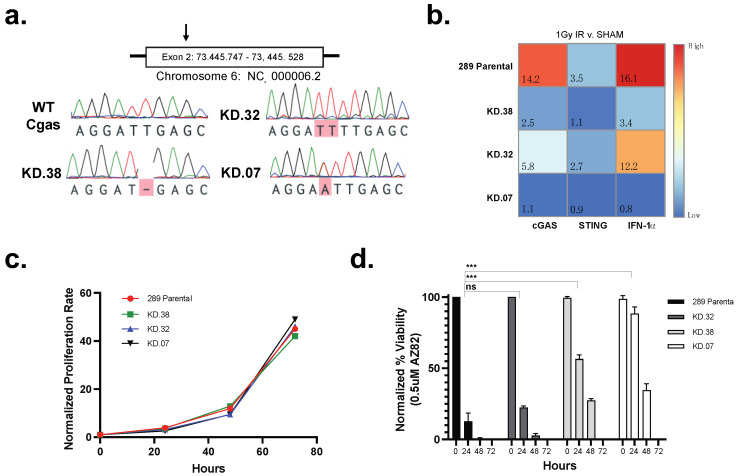
(**a**) Sanger sequencing results from parental 289 cells and three knock-down clones (KD.38, KD.32, KD.07) with CRISPR edited cGAS at exon 2. Bases highlighted in red indicate sequence variation in comparison to 289 parental cGAS. (**b**) Fold-change gene expression of cGAS, STING, and IFN-α1) in 289 parental cells and three clones (KD.38, KD.32, KD.07). Samples were analyzed 24 h after exposure to 1 Gy X-irradiation or sham treatment. Heatmap represents the relative fold-change of irradiated samples normalized to sham samples. (**c**) Proliferation rate of parental 289 cells and three knock-down clones (KD.38, KD.32, KD.07). Cells were stained with trypan blue and counted with a hemocytometer at 0, 24, 48, and 72 h. (mean ± SEM, n = 3 experiments) (**d**) Normalized % viability of parental 289 cells and three knock-down clones (KD.38, KD.32, KD.07) treated with 0.5 μM AZ82. Cells were stained with trypan blue and counted with a hemocytometer at 0, 24, 48, and 72 h. (mean ± SEM, n = 3 experiments, unpaired two-tailed *t*-test for 24 h time point, *** *p* < 0.001, ns = not significant).

## Data Availability

Not applicable.

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
