# Peer review of "Centrosome Amplification Is a Potential Molecular Target in Paediatric Acute Lymphoblastic Leukemia"

_cancers, 2022, doi:10.3390/cancers15010154_

Round 1

Reviewer 1 Report

Manuscript „Centrosome amplification is an operative molecular target in paediatric acute lymphoblastic leukemia“ describes very interesting study and reveals tumor-cell specific mitotic behavior that can be used as a potential therapeutic target.

However, there are few points that need attention:

Methods used for results written in section: 3.4. cGAS is necessary for 289 B-ALL cell responses to CCI („To evaluate the necessity for cGAS in the response of 289 B-ALL cells to CCIs, we 355 used CRISPR-Cas 9 to generate three cGAS knock down (KD) clones targeting exon 2 356 (KD.38, KD. 32, and KD.07) selecting one clone showing a parental phenotype (KD.32), 357 and two clones showing a null (KD.07) or near null (KD.38) phenotype. After detecting 358 gene alterations with Sanger sequencing (Figure 4a)…“,) are not described in section Methods.

Section 2.6. Reagents, antibodies, and drugs is redundant – it is easier to read and understand methods if reagents are just given in brackets when mentioning them in methods description.

Figures 1. and 2. show a lot of data and it would be easier to follow the logic of the Results if these figures are divided in more separate pictures that would in this way also be more visible.

I'd suggest to avoid title as a sentence and to avoid „operative“, using instead maybe „potential“ or something similar.

In section Statistic a bit more data could be useful – how was which of data sets analyzed?

I would also suggest to avoid repetition of introduction and aims in results section.

It would be beneficial if data gained on mouse and human cells are more clearly divided in comments in section Discussion.

Reviewer 2 Report

Manuscript No.: cancers-2099293.

Recommendation: Minor revision.

Comments:

The article entitled “Centrosome amplification is an operative molecular target in paediatric acute lymphoblastic leukemia” This paper looks scientifically sound, and interesting to the readers of cancers journal. After addressing the few recommendations made in this article. The manuscript may be accepted for publication in this journal. A few minor corrections are suggested:

1.      The abstract is too long; I ask the authors to shorten it and leave only the most important information.

2.      To put the study in its state of the art, the authors must update the Introduction section, by recent works found in the literature. Introduction section should be revised.  Some new references which should be added.

3.      Why is the choice of the method employed in this study?

4.      How the author do chose different proteins in this study? Explain more and compare the results found with these found in the literature.

5.      The manuscript should be thoroughly scrutinized for punctuations/grammatical errors.

Remark: Accept with minor revision

Reviewer 3 Report

Major comments:

1. This study lacks transcriptome-wide data. In the post TCGA-times qPCR data on their own are insufficient.

2. Why studies like as presented in Fig. 1 are done in mouse and not in human cells?

Minor comments:

1. All abbreviations should be defined at their first time use and used then consistently, i.e. no repeated definitions. This applies also to the Abstract and gene/protein names.

2. There are a number of details where the technical manuscript preparations seems to be sloppy. Please improve.

Round 2

Reviewer 3 Report

Gene and protein name abbreviations are still not defined, please improve.
